# PBR Clutter Suppression Algorithm Based on Dilation Morphology of Non-Uniform Grid

**Qian Zhu [1], Tao Li [2], Jiameng Pan [1] and Qinglong Bao [1,*]**

[1]    Science and Technology on Automatic Target Recognition Laboratory, National University of Defense Technology, Changsha 410072, China; zq1142006@126.com (Q.Z.); 3090100472@zju.edu.cn (J.P.)

[2]    Artificial Intelligence Research Center (AIRC), National Innovation Institute of Defense Technology (NIIDT), Beijing 100071, China; litao_nudt@163.com

*    Correspondence: cbpest@163.com; Tel.: +86-132-7241-5200

**Abstract:** Many new challenges are faced by the PBR (passive bi-static radar) employing non-cooperative radar illuminators. After the CFAR (constant false alarm) processor, the appearance of the amount of false alarm clutter points impacts the following tracing performance. To enhance the PBR tracing performance, we consider to reduce these clutter points before target tracing as soon as possible. In this paper, we propose a PBR clutter suppression algorithm based on dilation morphology of non-uniform grid. Firstly, we construct the non-uniform polar grid based on the acquisition geometry of PBR. Then, with the help of the grid platform, we separate the false alarm clutter points based on the dilation morphology. To efficiently operate the algorithm, we build up its parallel iteration scheme. To verify the performance of the proposed algorithm, we utilize both simulated data and field data to do the experiment. Experimental results show that the algorithm can effectively suppress most of the clutter points. Besides, we respectively combine the proposed suppression algorithm with two typical tracking algorithms to test the performance. Experimental results reveal that the compound tracing algorithm outperforms the traditional one. It can enhance the PBR tracing performance, reduce the occurrence probability of false tracks and meanwhile save time.

**Keywords:** PBR (passive bistatic radar); clutter suppression; non-uniform grid; dilation morphology

## 1. Introduction

Passive radar, employing non-cooperative illuminators, has attracted increasing interests in recent years [1–12]. It has many obvious advantages over traditional active radars, such as low-cost, feasibility of various illuminators and immune to the anti-radiation missiles [1]. In existing literature, illuminators of opportunity for passive radar are generally categorized into four groups: broadcast signals (DVB-T, FM, DAB, etc.) [3,4], mobile communication signals (Wi-Fi, GSM (Global System for Mobile communication), LTE (Long Term Evolution), etc.) [6,7], geolocalization signals (GNSS (Global Navigation Satellite System), GPS (Global Positioning System)) [8–10] and radar signals [11,12]. Most of researches focus on passive systems employing the first three types of illuminators, while the literature on passive systems employing non-cooperative radar signal as illuminator is rare because of the difficulties in signal processing. In this paper, we explore the research based on the PBR (passive bi-static radar) employing non-cooperative radar illuminators.

The operation geometry of the PBR system is illustrated in Figure 1. The system consists of two channels: echo channel and reference channel. The former is designed for receiving scattered wave, and the latter is for direct wave. The uncooperative transmitter is generally equipped with phased array. Compared to the traditional mechanical scanning radar, the phased array radar has flexible multi-beam scanning and various beam dwell times. It can achieve search and tracing simultaneously.

Nowadays, many modern radars are equipped with phased array for detection. It is meaningful to exploit the phased array radar signal as the illuminator. The beam scan of the uncooperative illuminator is agile and flexible with unknown purpose. Since it is hard to predict and track its rapid changing beam steering, we choose to adopt multi-beam forming simultaneously covering the surveillance range [11] to realize space synchronization. Besides, to enhance the ability of anti-jamming and the detection probability, the uncooperative radar usually transmits the signal that is agile in frequency, PW (pulse width), BW (band width) and PRI (pulse recurrence interval). Based on the characteristics of PBR above, it faces many new challenges.

- The space synchronization accuracy is not as good as the traditional radar, resulting in the decreased SNR (signal noise ration) and the poor location precision.
- Simultaneous multi-beam forming leads to the redundant data being increased.
- The reference channel is not ideally compatible to the echo channel due to the multipath and the minor difference of antenna performance. The performance of the following pulse compression degrades.
- Due to the agility of the illuminator parameters, the number of the pulses utilized for detection is less. Besides, the scattered wave of the target depends on the opportunity of the beam steering. Thus, the valid data rate is decreased.
- Since the illuminator parameters are agile pulse by pulse, it is hard to adopt coherent integration to suppress clutter like traditional radar.
- Low SNR calls for low threshold during CFAR (constant false alarm), that is to increase the detection rate, whereas the false-alarm rate increases correspondingly.

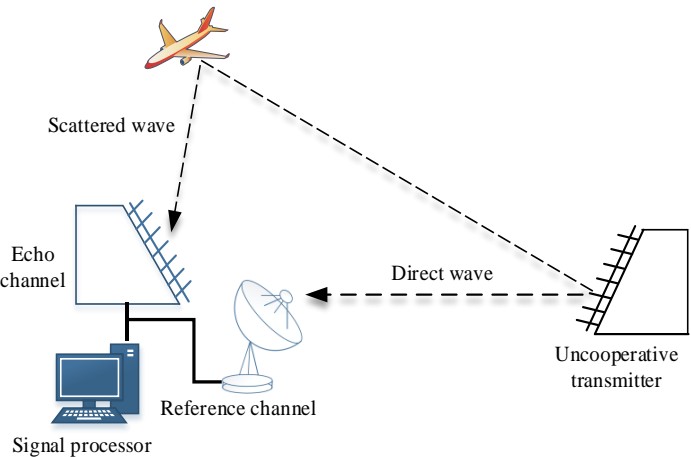

**Figure 1.** Geometry of PBR utilizing uncooperative radar signal as transmitter.

Thus, after pulse compression and CFAR processor, the difficulties during target tracking can be concluded into four points. That is breaking tracks, amounts of false-alarm clutter data, random interval between adjacent tracing points and huge computation.

To enhance the PBR tracking performance, we consider to reduce the false-alarm clutter points before tracking as soon as possible. The existing clutter suppression algorithms for passive radar are aimed at the direct wave interference and ground clutter. In addition, they mainly focus on the spatial domain, the temporal domain and the sub-carrier domain. In spatial domain, there is ABF (adaptive beamforming) and its extension version [13,14]. In temporal domain, researchers propose many adaptive filter algorithms applied on PBR, such as LMS (least mean square) [15], fast-block LMS [16], GANF (generalized adaptive notch filter) [17] and so on. Adaptive filter is of low convergence speed; however, ECA [18] (extensive cancellation algorithm) covers its deficiency. In recent years, many algorithms around ECA have been proposed, such as ECA-S (ECA-sliding) [19],

ECA-ES (ECA-expectation simplified) [20] and so on. In sub-carrier domain, algorithms only work in orthogonal frequency-division multiplexing-based PBR, such as RLS-C (recursive least square by sub-carrier) [21], ECA-C (ECA by carrier) [22], ECA-CD (ECA by carrier and doppler shift) [23] and so on.

In addition to the direct wave interference and ground clutter, the radar illuminator-based PBR is also influenced by the false-alarm clutter during processing, as analyzed above. Rare literatures discuss the false-alarm clutter suppression algorithm in spatial–temporal domain before tracing. In this paper, we aim to put forward a PBR false-alarm clutter suppression algorithm. To make a low budget solution, we resort to the grid-based method so that we avoid calculating point-to-point Euclidean distance. In [24], grid-based DBSCAN is proposed for clustering objects in radar data. The method is not specially designed for PBR and its model is simple. In [25], ENM (ellipsoid norm method) is proposed to promise optimal result in passive multi-static location. It focuses on finding the nearest grid point in grid-based method. However, it only works in the noise-free scenario and is not suitable for dense clutter environment. Thus, based on the acquisition geometry of PBR, we firstly propose a non-uniform polar grid construction method. In addition, with the help of the grid platform, a false-alarm clutter separation method is proposed based on dilation morphology. The combination of these two steps is the whole algorithm we proposed in this paper.

The remaining part is organized as follows. Section 2 illustrates the geometrical relationship of PBR and describes the proposed non-uniform polar grid construction method. Section 3 describes the grid-based clutter suppression method and its parallel iteration scheme. Section 4 describes both simulated data experiment results and field data experiment results. Section 5 is the conclusion.

## 2. Non-Uniform Polar Grid Construction for PBR

According to the bi-static radar position principle, Figure 2 demonstrates the geometrical relationship between target, transmitter and receiver. $R_r$ stands for the range of the target, that is the distance between target and receiver. $R_t$ is the distance between target and transmitter. L stands for the baseline range between transmitter and receiver. $\beta$ stands for the bi-static angle, that is the intersection angle between the line from receiver to target and the line from transmitter to target. $\theta_r$ stands for the azimuth angle, that is the supplementary angle between the line from receiver to transmitter and the line from receiver to target.

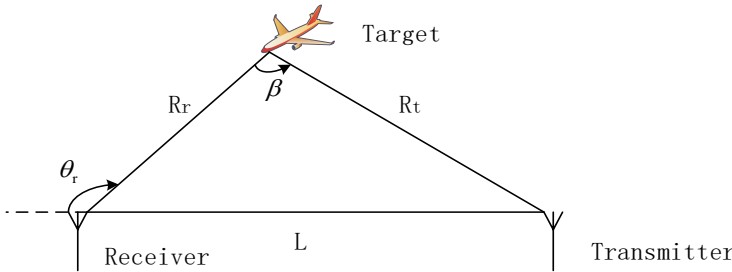

**Figure 2.** The geometrical relationship of PBR.

To locate the target, we should find its azimuth and range. The azimuth can be directly measured through the passive bi-static radar system. However, the range can only be calculated through the bi-static range sum. Suppose $R_S$ is the bi-static range sum. Thus,

$$R_s = R_t + R_r \tag{1}$$

Referring to the cosine law, the relationships between $R_r$ and $R_s$ can be derived into Equations (2) and (3).

$$R_r = (R_s^2 - L^2)/(2(R_s + Lcos\theta_r)) \tag{2}$$

$$R_s = R_r + \sqrt{R_r^2 + L^2 + 2R_rLcos(\theta_r)} \tag{3}$$

From Equation (2), we can find that the mono-static range $R_r$ is a non-linear function of the bi-static range sum $R_S$ and the azimuth angle $\theta_r$.

In passive bi-static radar system, firstly, achieve the time synchronization of the echo channel with the help of the reference channel. Then, measure the difference between $R_S$ and L through the time delay of echo. Since L is fixed, $R_S$ can be directly measured from echo channel. As $R_S$ and $\theta_r$ are direct measurements, the measuring and detecting error transfer to $R_r$ by Equation (2) at last. Due to non-linear relationship between $R_S$ and $R_r$, it is necessary for us to construct a non-uniform grid for the latter clutter suppression processing.

At first, we derive the first order Taylor series expansion of Equation (2) at an arbitrary position $(R_{s0}, \theta_0)$, shown in Equation (4).

$$
\begin{aligned}
R_r(R_{s0} &+ \Delta r, \theta_0 + \Delta\theta) \\
&= R_r(R_{s0}, \theta_0) + \Delta r \frac{\partial R_r(R_s, \theta_0)}{\partial R_s}|_{R_s = R_{s0}} + \Delta\theta \frac{\partial R_r(R_{s0}, \theta)}{\partial \theta}|_{\theta = \theta_0} + o(\Delta r, \Delta\theta)
\end{aligned}
\tag{4}
$$

Thus, the transfer error of the $R_r$ caused by $R_S$ and $\theta$ at the position $M(R_{s0}, \theta_0)$ is derived below.

$$
\begin{aligned}
\Delta R_r &= R_r(R_{s0} + \Delta r, \theta_0 + \Delta\theta) - R_r(R_{s0}, \theta_0) \\
&= \Delta r \frac{\partial R_r(R_s, \theta)}{\partial R_s}|_{R_s = R_{s0}, \theta = \theta_0} + \Delta\theta \frac{\partial R_r(R_s, \theta)}{\partial \theta}|_{R_s = R_{s0}, \theta = \theta_0} + o(\Delta r, \Delta\theta) \\
&= \Delta r \frac{R_{s0}^2 + L^2 + 2R_{s0}Lcos(\theta_0)}{2(R_{s0} + Lcos(\theta_0))^2} + \Delta\theta \frac{(R_{s0}^2 - L^2)Lsin(\theta_0)}{2(R_{s0} + Lcos(\theta_0))^2} + o(\Delta r, \Delta\theta)
\end{aligned}
\tag{5}
$$

Assume $\rho_1(R_{s0}, \theta_0) = \frac{R_{s0}^2 + L^2 + 2R_{s0}Lcos(\theta_0)}{2(R_{s0} + Lcos(\theta_0))^2}$, $\rho_2(R_{s0}, \theta_0) = \frac{(R_{s0}^2 - L^2)Lsin(\theta_0)}{2(R_{s0} + Lcos(\theta_0))^2}$.

Then, omit the Peano remainder term $o(\Delta r, \Delta\theta)$.

$$
\Delta R_r(\Delta r, \Delta\theta)|_{R_s = R_{s0}, \theta = \theta_0} \approx \rho_1(R_{s0}, \theta_0)\Delta r + \rho_2(R_{s0}, \theta_0)\Delta\theta
\tag{6}
$$

From Equation (6), we can find that when $\Delta r$ and $\Delta\theta$ are fixed, the transfer error changes with the position $(R_{s0}, \theta_0)$. To facilitate following operation, we divide the detection coverage into grids based on the transfer error expansion and project the processing data into grids.

Assume the bi-static range error $\Delta r$ and the azimuth angle error $\Delta\theta$ obey the Gaussian distribution with zero-mean, as shown below.

$$
\Delta r \sim N(0, \sigma_r^2); \Delta\theta \sim N(0, \sigma_\theta^2)
\tag{7}
$$

In general, $\sigma_r$, the standard deviation of $\Delta r$, mostly relates to the range resolution. In addition $\sigma_\theta$, the standard deviation of $\Delta\theta$, mostly relates to the space synchronization accuracy and the array error. We suppose them as known constant. Further discussion about them is not included in this paper.

Since $\Delta R_r$ is the linear function of $\Delta r$ and $\Delta\theta$, $\Delta R_r$ is also obey the Gaussian distribution with zero-mean, as shown below.

$$
\Delta R_r \sim N(0, \rho_1^2(R_{s0}, \theta_0)\sigma_r^2 + \rho_2^2(R_{s0}, \theta_0)\sigma_\theta^2)
\tag{8}
$$

There is a $3\sigma$ principle of Gaussian distribution in the probability theory. In Gaussian distribution, the probability of the data distributing in the range $(\mu - 3\sigma, \mu + 3\sigma)$ is 99.74%. Where $\mu$ is the mean value of sample set. $\sigma$ is the standard deviation of database.

Based on the analysis above, the grid spacing is designed shown in Equations (9) and (10).

$$
\delta_r(R_{s0}, \theta_0) = 3\sqrt{\rho_1^2(R_{s0}, \theta_0)\sigma_r^2 + \rho_2^2(R_{s0}, \theta_0)\sigma_\theta^2}
\tag{9}
$$

$$
\delta_\theta(R_{s0}, \theta_0) = 3\sigma_\theta
\tag{10}
$$

where $\delta_r(R_{s0}, \theta_0)$ and $\delta_\theta(R_{s0}, \theta_0)$ are the grid spacing at the position $(R_{s0}, \theta_0)$ in range dimension and angular dimension respectively.

As the $\delta_\theta$ is uncorrelated with the position $(R_{s0}, \theta_0)$, the grid is divided evenly in the angular dimension. However, in range dimension, the pace will be iteratively calculated based on the present position. Assume the observing scope is from $\theta_0$ to $\theta_{max}$ and from $r_0$ to $r_{max}$. The polar mesh grid calculation step is shown in Table 1.

**Table 1.** The polar mesh grid calculation steps.

| **1. Calculate angular coordinate** |
| --- |
| The angular coordinate set is $\Theta = \left\{\theta_k \vert \theta_{k+1} - \theta_k = 3\sigma_\theta, k = 0\ldots N, N = \left[\frac{\theta_{max}-\theta_0}{3\sigma_\theta}\right]\right\}$. Where $[\cdot]$ is the symbol of round down, and N is the mesh counts in angular dimension. |
| **2. Aiming at each angular coordinate $\theta_k$ in $\Theta$, iteratively calculate the grid division in range dimension.** |
| For $\theta_k, k = 0\ldots N$<br>Initialization: $i = 0$, $R_i = r_0$;<br>Iteration: $Rs_i = R_i + \sqrt{R_i^2 + L^2 + 2R_i L cos(\theta_k)}$;<br>$\rho_1(R_{si}, \theta_k) = \frac{R_{si}^2 + L^2 + 2R_{si}Lcos(\theta_k)}{2(R_{si}+Lcos(\theta_k))^2}$, $\rho_2(R_{si}, \theta_k) = \frac{(R_{si}^2-L^2)Lsin(\theta_k)}{2(R_{si}+Lcos(\theta_k))^2}$.<br>$R_{i+1} = R_i + 3\sqrt{\rho_1^2(R_{si}, \theta_k)\sigma_r^2 + \rho_2^2(R_{si}, \theta_k)\sigma_\theta^2}$;<br>$i = i + 1$;<br>Terminate when $R_i > r_{max}$.<br>$N_k = i$. $N_k$ is the mesh counts in range dimension for $\theta_k$.<br>The range coordinate set is $\Lambda = \left\{R_{i,\theta_k} \vert i = 0\ldots N_k, \theta_k \in \Theta\right\}$. |

Referring to the proposed polar grid calculation algorithm, we calculate and make an example map of the polar grid in Figure 3. Set the observing scope 100° to 160° in azimuth and 20 km to 100 km in range. Blue lines denote the grid division in angular dimension, while red lines stand for the grid division in range dimension. It is obvious that closer to the baseline angle 180°, the grid size is bigger.

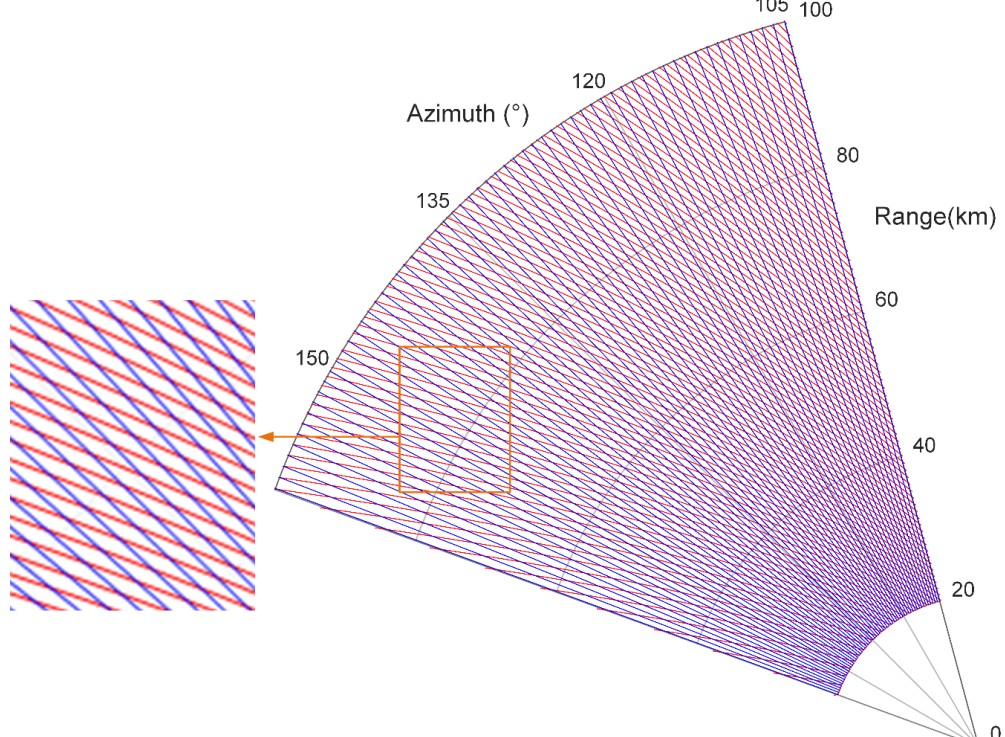

**Figure 3.** An example map of the proposed polar grid.

## 3. Separate False Alarm Clutter from Data

We utilize the location dependency of target track within consecutive frames to separate false alarm clutter from data. To reduce the computation load, the method in this section is proposed based on the grid constructed in Section 2. More specifically, in this section, we resort to morphology, usually applied in image processing, to achieve clutter separation.

The process is divided into three steps. Firstly, mark the point on the polar grid according to its position measured. Secondly, separate false alarm clutter point from data based on morphology algorithm. Thirdly, iteratively complete the operation frame by frame.

### 3.1. Mark the Point on Grid

After the CFAR processor and the Direction of Angle (DoA) estimation, we obtain the point data carrying with its own location information, namely range and azimuth angle. The first step of the separation is to mark those points on the grid constructed in Section 2 based on their locations. As the grid is uniform in angular dimension, to facilitate the calculation, we operate in order from azimuth angle to range. Assume $n_a$, $n_r$ as the grid index of the point in angular dimension and range dimension respectively. r and $\alpha$ refer to its range and azimuth angle. The calculation method is described in Equations (11) and (12).

$$n_a = \left[ \frac{\alpha}{\Delta\theta} \right] + 1 \tag{11}$$

$$n_r = \{ind | net(ind, n_a) \leq r < net(ind + 1, n_a)\} \tag{12}$$

where $[\cdot]$ is the symbol of round down. $\Delta\theta$ is the angular spacing of the grid. *net* is the grid coordinate matrix in accord with the range coordinate set $\Lambda$ which is calculated in Section 2. The number of columns in *net* is the same as the number of elements in the angular coordinate set $\Theta$ where each column vector is the range division corresponding to each angular value. According to the range and angular index calculated above, mark the point on a matrix $A$ with the same size of *net*.

### 3.2. Separate False Alarm Clutter from Data Based on the Dilation Morphology

In general, the true points of the target track have the feature of location dependency among several consecutive frames, whereas the false alarm clutter points are relatively isolated. Assume $F_n$ is the object frame. $\Psi = \{F_{n-k} \ldots F_{n-1}, F_{n+1} \ldots F_{n+k}\}$ is the group of the reference frames, before and after several frames of $F_n$. Where k is the half number of the reference frames. Firstly, mark the points of the frame in group $\Psi$ one by one. Obtain the mark-matrix $A_{n-k} \ldots A_{n-1}, A_{n+1} \ldots A_{n+k}$ respectively, and compose them into a new mark-matrix $\Gamma_n$, shown in Equation (13). Meanwhile, mark the points of the object frame on the matrix $A_n$.

$$\Gamma_n = A_{n-k} \cup \ldots \cup A_{n-1} \cup A_{n+1} \cup \ldots \cup A_{n+k} \tag{13}$$

To facilitate following iteration calculation, we replace (13) by another more specific operation, described in Equation (14). Where $\mathfrak{Bm}(\cdot)$ is the symbol of binarization.

$$\Gamma_n = \mathfrak{Bm}(A_{n-k} + \cdots + A_{n-1} + A_{n+1} + \cdots + A_{n+k}) \tag{14}$$

Next, to ensure the target points in the neighborhood of the points from reference frames, we do morphological dilation on $\Gamma_n$ with a rectangular structural element B. The size of B depends on the coarse estimation of the target's move range. The dilation result matrix marks the neighborhood area of the points from reference frames. $M_n$ in Equation (15) is the dot product of the dilation result and the object mark-matrix $A_n$.

$$M_n = (\Gamma_n \oplus B) \cdot *A_n \tag{15}$$

where ⊕ is the symbol of dilation. ∗ is the symbol of the dot product. The matrix $M_n$ stands for the final marked area of screened data for the frame $F_n$. To facilitate realization, the dilation of binary matrix can be expressed by binarization after convolution. So Equation (15) can be rewritten as

$$M_n = \mathfrak{Bm}(\Gamma_n \otimes B) \cdot {*}A_n \tag{16}$$

where ⊗ is the symbol of convolution.

The processing progress is illustrated in Figure 4.

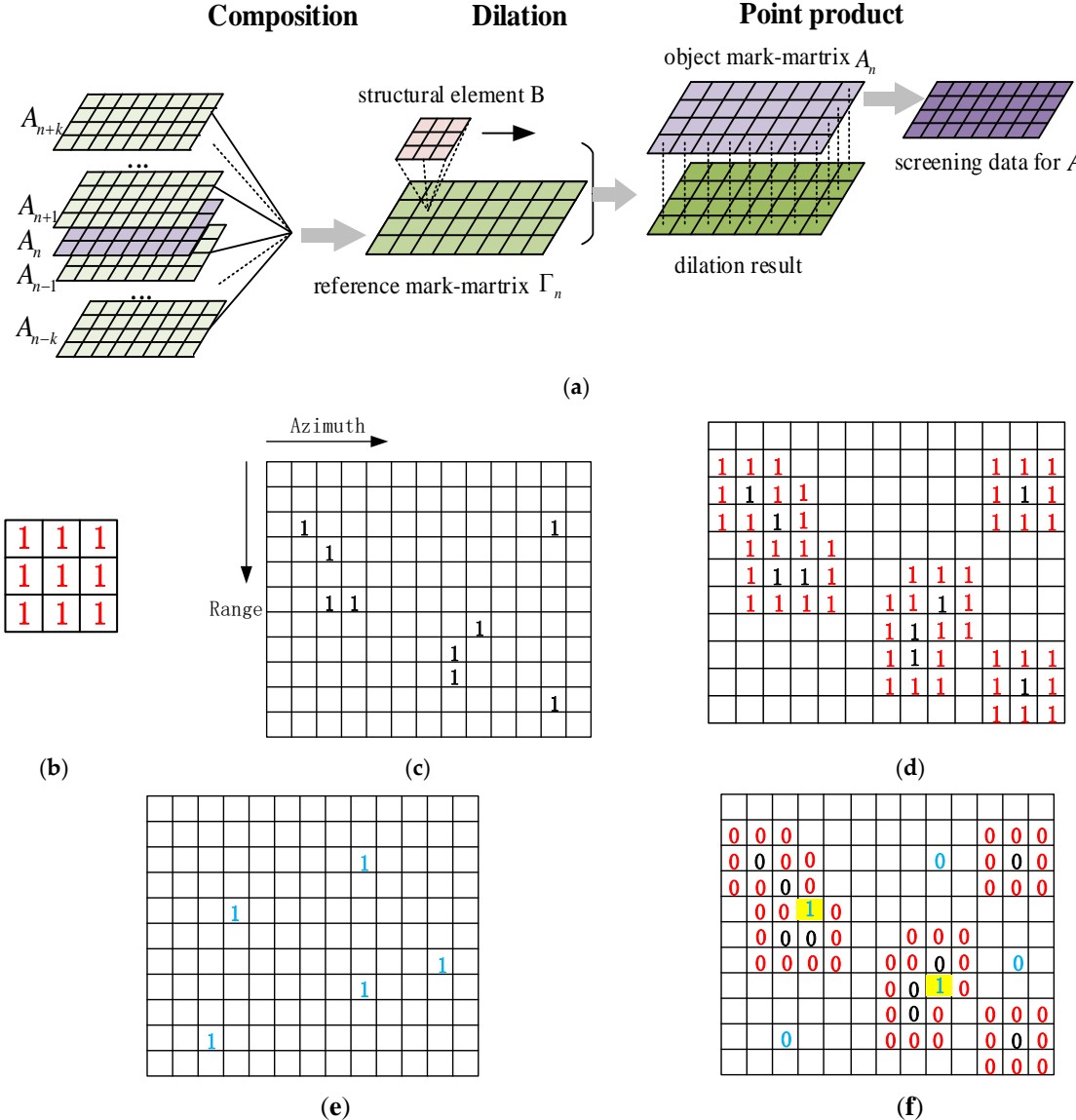

**Figure 4.** The sketch map of processing progress: (**a**) general view; (**b**) structural element B; (**c**) reference mark-matrix $\Gamma_n$; (**d**) dilation result; (**e**) object mark-matrix $A_n$; (**f**) screened data for $A_n$.

We choose a simple case to clearly illustrate how the proposed algorithm operates. Figure 4a indicates the general view of the operation. Figure 4b–f shows the processing result in each step. After the composition of reference frames, dilation and the point product with object frame, we can obtain the screened data, where the clutter points are filtered out. In this case, Structural element B is a 3 × 3 square matrix, as shown in Figure 4b. Figure 4c shows the reference mark-matrix $\Gamma_n$. Figure 4d shows the dilation result of the B and $\Gamma_n$. Figure 4e is the object mark-matrix $A_n$ with five suspected

areas. Figure 4f shows the result, the screened data for $A_n$. Obviously, the points in two areas of yellow background are retained, yet rest of them are suspected as clutters and filtered out.

It is noteworthy that since the transfer error changes with the point location, the real size and shape of the structural element for dilation is not fixed in fact. It changes along with the location of the suspected point. However, as the processing data has been abstracted through the non-uniform polar grid, the change of the structural element does not involve in this section. The left part of Figure 5 is the partial enlarged map of Figure 3. There are two structural elements, corresponding to point A and B, with different shape and size. Both are projected into the same element for convenience during processing.

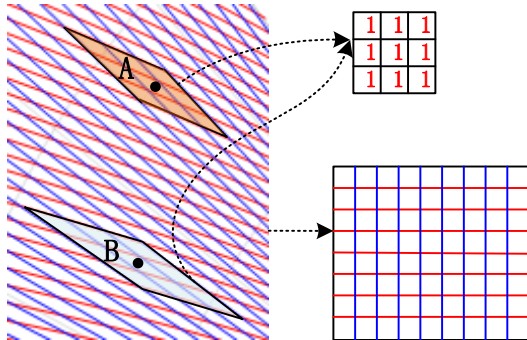

**Figure 5.** The sketch map of the grid projection.

### 3.3. Iteratively Calculation Frame by Frame

To efficiently operate the algorithm, we further explore its iterative and parallel calculating scheme. Referring to Equation (14), we can derive the reference mark-matrix $\Gamma_{n+1}$:

$$
\begin{aligned}
\Gamma_{n+1} &= \mathfrak{Bm}(A_{n+1-k} + \cdots + A_n + A_{n+2} + \cdots + A_{n+1+k}) \\
&= \Gamma_n + \mathfrak{Bm}(A_n + A_{n+1+k} - A_{n+1} - A_{n-k}) \\
&= \Gamma_n + \mathfrak{Bm}(A_n - A_{n-k}) + \mathfrak{Bm}(A_{n+1+k} - A_{n+1})
\end{aligned}
\tag{17}
$$

Assume

$$
\Psi A_n = \begin{cases} \mathfrak{Bm}(A_n - A_{n-k}) & n > k \\ \mathfrak{Bm}(A_n) & n \le k \end{cases}
\tag{18}
$$

Thus, Equation (17) will be derived into Equation (19).

$$
\Gamma_{n+1} = \Gamma_n + \Psi A_n + \Psi A_{n+1+k}
\tag{19}
$$

where k is the half number of the reference frames. When $n = 1$, $\Gamma_1 = \mathfrak{Bm}(A_2 + \cdots + A_{1+k})$.

It can be observed that Equations (11), (12), (16), (18) and (19) are relatively independent of calculation. To promote the efficiency of the algorithm, we split the whole process into two parts for parallel calculation and build up an intermediate database to link them together. One is frame processing part, and the other is interframe processing part. Both are designed to operate in parallel. Based on the above analysis, the clutter separation algorithm is described in below chart, shown in Figure 6.

The data frame $F_i$ flown from the CFAR processor is input in this system. $i$ is the index of the current frame flowing in. When $i > k$, the interframe processing part starts operating. $n$ is the index of the current processing frame, and is independent of the index $i$. In frame processing part, mark the data of $F_i$ on the grids and calculate $\Psi A_i$. Save $A_i$ and $\Psi A_i$ into the intermediate database called by clutter separation in interframe processing part. Finally, iteratively compute and output the mark matrix $M_n$ in the interframe processing part.

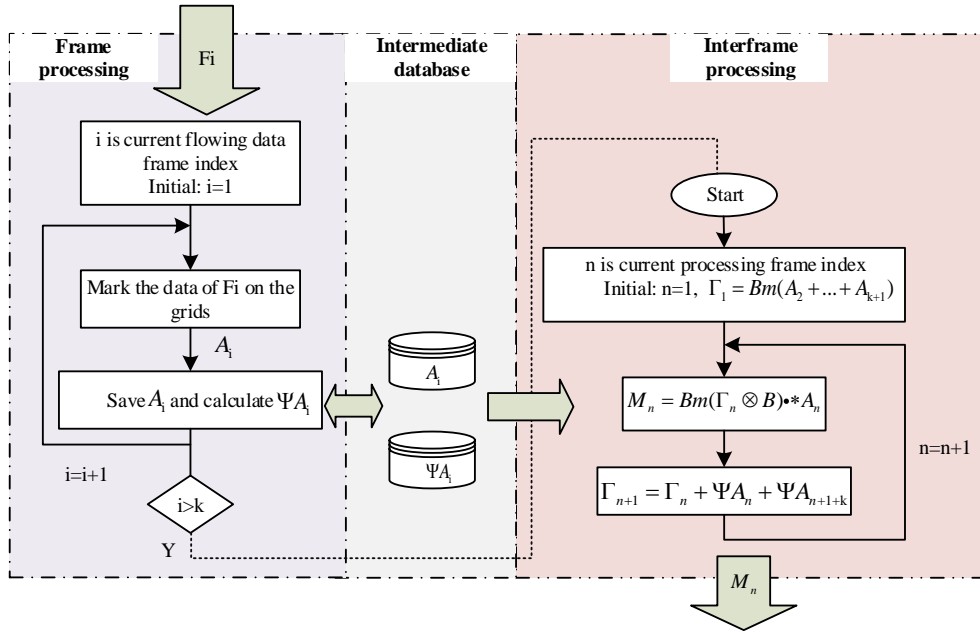

**Figure 6.** The flow chart of the clutter separation algorithm.

## 4. Experiment result and Analysis

In this section, the proposed algorithm is tested on a simulated scenario in Section 4.1 and a real scenario in Section 4.2. The computer configuration for experiment: Inter(R) Core(TM) i7-4790 CPU @ 3.60 GHz. RAM: 16.0 GB. All operations in this section run on Matlab R2018a.

### 4.1. Testing by Simulated Data

#### 4.1.1. Scenario for Simulation

The surveillance scope is set from 60° to 170°. There are five targets moving with constant velocity in the scope. Table 2 lists the track settings of five targets. Plot these target tracks in polar coordinates, as Figure 7a shows.

**Table 2.** The track settings of five targets.

|  | Start Position (km, degree) in Polar Coordinates | Start Position (km) in Cartesian Coordinates | Track Slope | Track Intercept (km) |
|---|---|---|---|---|
| Target 1 | (86.023,144.5) | (−70, 50) | 5 | 60 |
| Target 2 | (70.456,96.5) | (−8, 70) | 10 | 100 |
| Target 3 | (80.623,82.9) | (10, 80) | −3 | 10 |
| Target 4 | (76.158,113.2) | (−30, 70) | −10 | 80 |
| Target 5 | (70.711,135) | (−50, 50) | 30 | 55 |

Since the parameter of the illuminator is agile and various, PBR utilizes only part of pulses with specific aims for detection. To make closer to reality, the time interval between pulses utilized is not constant. The whole time length of simulated data is 50 s. The number of valid pulses is set as 667. The time interval between adjacent valid pulses is allocated randomly. Figure 8 shows the pulse interval allocation of simulated data. So, the detection result from the ideal echo of the target is not uniformly continues. Figure 7b demonstrates the target tracks points based on the pulse interval allocation with measurement error.

Besides, due to the flicker of target's RCS in PBR, target can only be detected from part of valid pulses. Figure 7c demonstrates the real target points detected. The signal to clutter ratio in this experiment is defined in Equation (20).

$$SCR = \log\left(\frac{N_{sig}}{N_{clu}}\right) \tag{20}$$

where $N_{sig}$ represents the mean number of the valid target detection points in each frame. $N_{clu}$ represents the mean number of the false alarm clutter points in each frame. Set SCR as −1.26 dB. Figure 7d shows the final detection result from CFAR, which is the simulated data for following experiment.

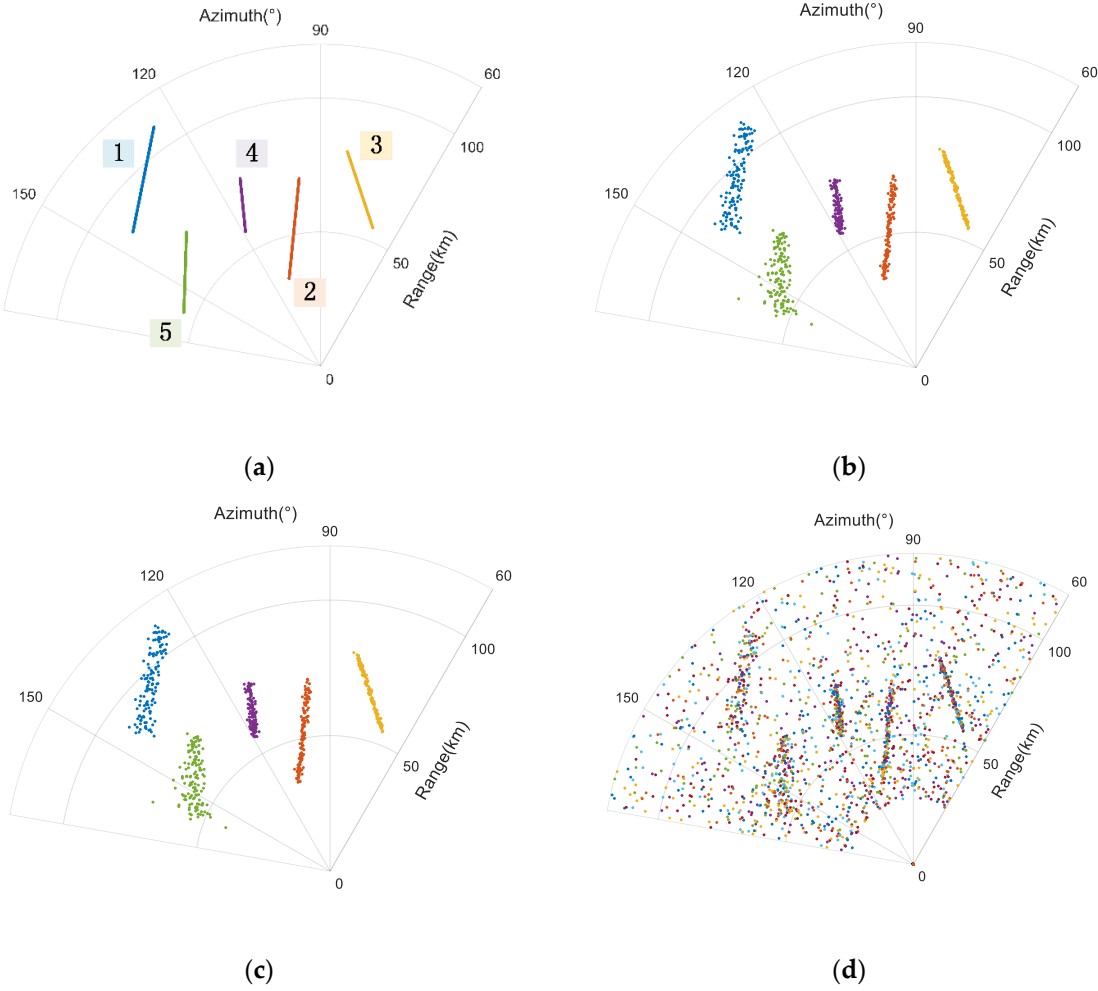

**Figure 7.** Production of the simulated data: (**a**) true tracks of five targets; (**b**) target tracks with measurement error; (**c**) target tracks detected; (**d**) detection results from CFAR.

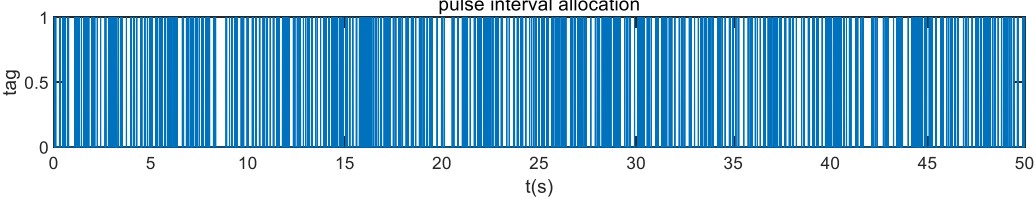

**Figure 8.** Pulse interval allocation.

### 4.1.2. The Clutter Suppression Performance Analysis

For grid construction, the standard deviation of range is set as 300 m according to the bandwidth of the illuminator. The standard deviation of angle is set as 0.3 according to the number of beams simultaneously covering the surveillance range. The range is from 20 km to 120 km. The angle scope is from 60° to 170°. The baseline range is 400 km. For frame operation, we categorize the simulated data into frame data by every 0.5 seconds. The structural element size is $3 \times 3$. The half number of the reference frames is set as 3. Set SCR as −1.26 dB. Figure 9 demonstrates the contrast before and after the suppression. In addition Table 3 shows three performance indexes of the suppression algorithm. The detection accuracy rate is the ratio of the number of correct target points extracted to the whole number of the correct target points. The false alarm decline rate is the ratio of the number of the false points extracted to the whole number of the clutter points set before. The miss detection rate is the ratio of the number of missing target points to the whole number of correct target points. From Table 3 and Figure 9, we can conclude that near 90% of clutter points are suppressed, while 97.45% of target points retain. To illustrate the algorithm performance comprehensively, change the SCR of the simulation scenario from −3 dB to 1 dB. In each SCR scenario, do Mont-Carlo experiment for 50 times and calculate the mean value of the performance indexes. We obtain following results, as Figure 10 shows. Red line stands for the correct detection rate. Blue line stands for the CFAR decline rate. Green line stands for the missing rate. Over 90% of target points can be extracted correctly in this algorithm. In addition when SCR is up to −2 dB, the number of false alarm points can be decline to the 30% of the original number. When SCR is up to −1 dB, over 90% of clutter points can be suppressed.

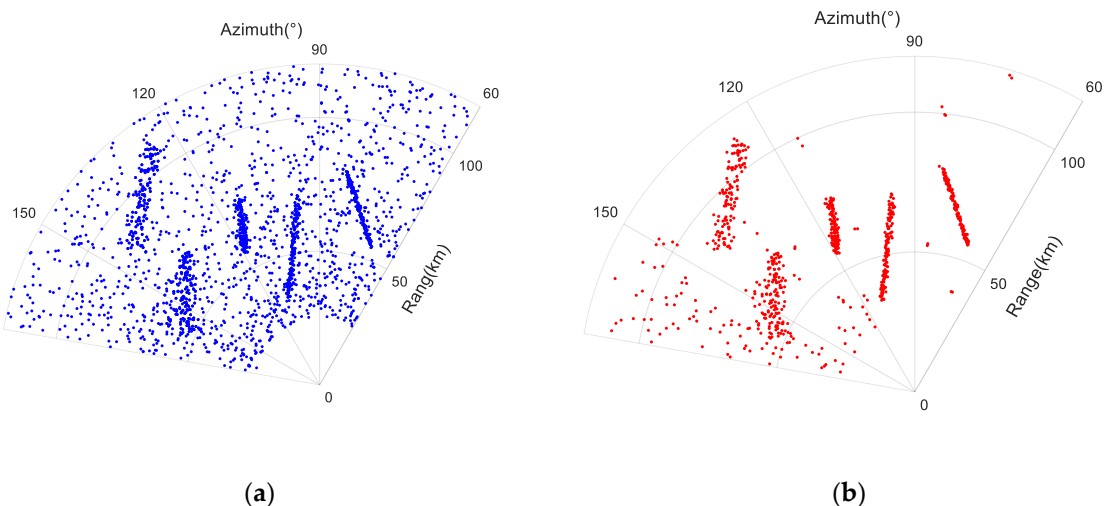

<div align="center">(<b>a</b>)                                                  (<b>b</b>)</div>

**Figure 9.** Contrast of before and after the suppression: (**a**) simulated data before suppression; (**b**) clutter suppression result.

**Table 3.** The performance indexes of the suppression algorithm.

| Detection Accuracy Rate | False Alarm Decline Rate | Miss Detection Rate |
|:---:|:---:|:---:|
| 97.45% | 10.24% | 2.55% |

### 4.1.3. Computation Analysis

When the uncooperative illuminator and the receiver of the PBR are located at fix sites, the non-uniform polar grid is fixed according to the acquisition geometry. The calculation of the grid is done in preprocess only once. So, the computation analysis of the grid construction is not involved in this section. In frame processing, assume the grid size is $m_1 * m_2$. The calculation of A for one processing point concludes $m_1 + m_2$ additions. The calculation of $\Psi A$ concludes $m_1 m_2$

additions. In interframe processing, assume the structural element size is $s_1 * s_2$. The calculation of $\Gamma$ concludes $2m_1 * m_2$ additions. The calculation of M needs $(s_1 * s_2 + 1) \cdot (m_1 * m_2)$ multiplications and $(s_1 * s_2 - 1) \cdot (m_1 * m_2)$ additions.

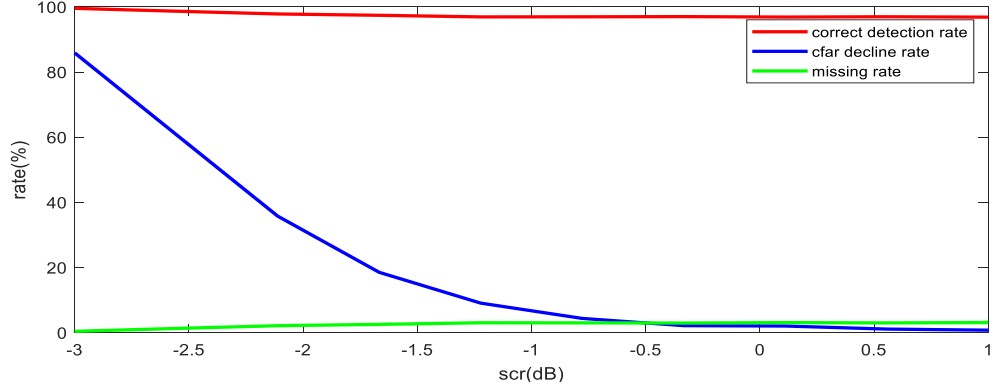

**Figure 10.** The change of the performance indexes along with SCR.

Suppose the frame number is k and the points number in each frame is n. So the whole process needs $k[n(m_1 + m_2) + (2 + s_1 * s_2) \cdot (m_1 * m_2)]$ additions and $k(s_1 * s_2 - 1) \cdot (m_1 * m_2)$ multiplications. That is $o(k * n)$ additions and $o(k)$ multiplications. Therefore, the addition times depends on the total number of points. In addition, the multiplication times only depends on the number of frames.

During tracing, the calculation amount relating to the number of clutter points is mostly caused by the Euclidean distance calculation between points. Assume M is the total number of points for tracing in each frame. In two consecutive frames, there are $M^2$ point pairs for processing. For each point pair, the calculation of the Euclidean distance needs 3 additions and 3 multiplications. Thus, the calculation amount of the Euclidean distance is $o(k * M^2)$ additions and $o(k * M^2)$ multiplications. Where k is the total number of frames.

Suppose our algorithm can suppress 90% clutter points, and this suppression process only consumes $o(k * M)$ additions and $o(k)$ multiplications. After suppression, the calculation amount of the Euclidean distance will descend to the 1% of the original. Thus, we pay low calculation amounts for reducing much more computation amounts of tracing.

### 4.1.4. Test the Performance Combining with Tracking Algorithm

We combine the proposed suppression algorithm with two typical tracking algorithms to test the performance. One is traditional NN TO-MHT algorithm (Nearest Neighbor Track Oriented-Multiple Hypothesis Tracking) [26], abbreviated as NN-MHT in this paper. The other is SNN-Kalman tracking algorithm (Suboptimal Nearest Neighbor - Kalman) [27], which is proposed aiming at multi-target tracking in non-cooperative passive system. To explicitly name the proposed algorithm, we name its abbreviation as MCSNG (Multi-frame Clutter Suppression based on Non-uniform Grid). Combine the clutter suppression with two tracking algorithms mentioned above respectively. For each frame, the tracking process follows with the clutter suppression in pipeline operation. Based on the difference of the tracking process, we name these compound algorithms as MCSNG-NN and MCSNG-SNN-K respectively. To make comparison, we utilize the original data without clutter suppression for tracing. Choose four indexes (total number of traces, mean trace length, maximum trace length, time consuming) to evaluate the tracing performance. Set the maximum velocity and the maximum accelerated velocity as 1000 m/s and 200 m/s$^2$ respectively. In Kalman filtering process, the maximum time period of blind prediction is set as 15 s.

Figure 11a,b illustrates the tracing result of NN-MHT and MCSNG-NN, respectively. In addition Table 4 lists the tracing result indexes of both. Utilize different colors to distinguish different tracks.

The number of false tracks produced from NN-MHT is 17, which is much more than MCSNG-NN. Besides, the time consuming of the MCSNG-NN is 76% less than the one of the NN-MHT.

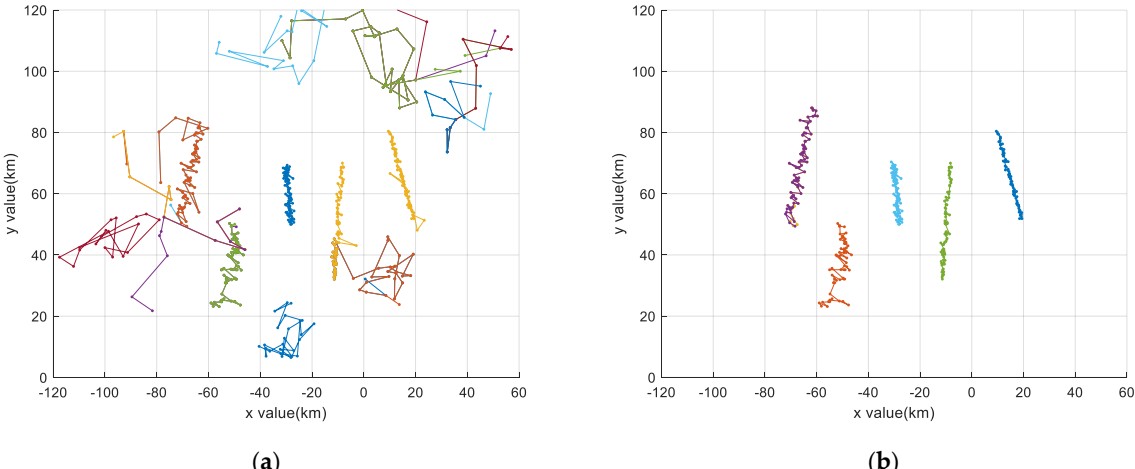

**Figure 11.** Tracing results comparison of both algorithms: (**a**) NN-MHT tracing result; (**b**) MCSNG-NN tracing result.

**Table 4.** The performance indexes of NN-MHT and MCSNG-NN.

|           | Total Number of Traces | Mean Trace Length | Max Trace Length | Time Consuming (s) |
| --------- | ---------------------- | ----------------- | ---------------- | ------------------ |
| NN-MHT    | 22                     | 39.13             | 89               | 2.99               |
| MCSNG-NN  | 6                      | 78.83             | 89               | 0.7                |

Figure 12a,b illustrates the valid tracing result of SNN-Kalman and MCSNG-SNN-K respectively. Comparing with five target tracks set before, we calculate the mean trace error of the valid tracing results from both algorithms, as Table 5 list. It is obvious that the tracking precision of MCSNG-SNN-K is higher than the one of SNN-Kalman. Since the time interval is not constant, the velocity estimation accuracy will be affected. However, the velocity estimation of MCSNG-SNN-K is closer to the true value than SNN-Kalman. Table 6 shows the index of the tracing result. It can be found that MCSNG-SNN-K is more efficiency and produces less false tracks than SNN-Kalman.

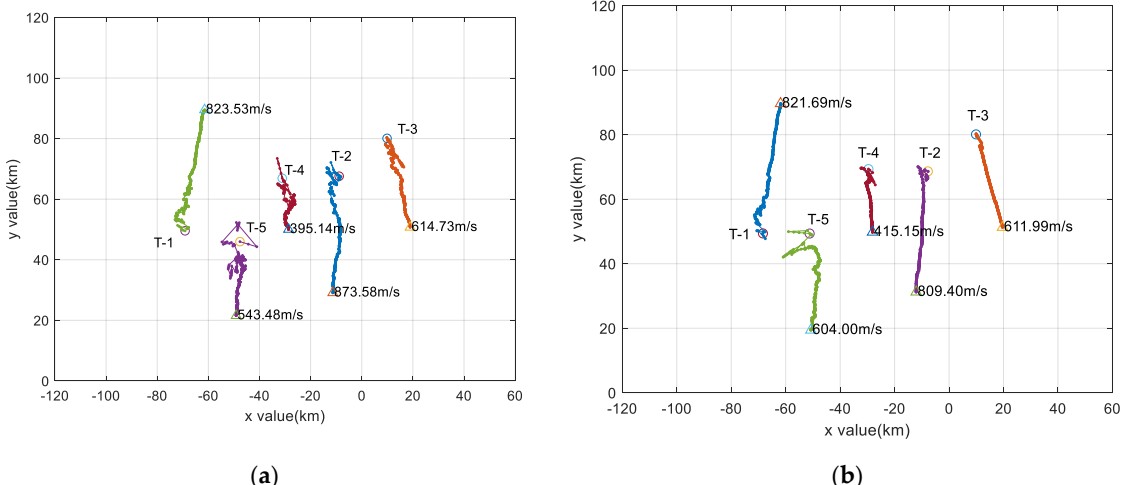

**Figure 12.** Valid tracing results comparison of both algorithms: (**a**) SNN-Kalman tracing result; (**b**) MCSNG-SNN-K tracing result.

**Table 5.** The comparison of five targets tracing results between SNN-Kalman and MCSNG-SNN-K.

| Track NO. | Mean Trace Error (m) | | Velocity (m/s) | | |
|---|---|---|---|---|---|
| | SNN-Kalman | MCSNG-SNN-K | SNN-Kalman | MCSNG-SNN-K | True Value |
| 1 | 548.87 | 434.56 | 823.5 | 821.7 | 800 |
| 2 | 1596.55 | 389.63 | 873.6 | 809.4 | 750 |
| 3 | 644.06 | 141.85 | 614.7 | 612.0 | 600 |
| 4 | 1286.59 | 283.97 | 395.1 | 415.2 | 400 |
| 5 | 2721.07 | 1840.33 | 543.5 | 604.0 | 600 |

**Table 6.** The performance indexes of SNN-Kalman and MCSNG-SNN-K.

| | Total Number of Traces | Mean Trace Length | Max Trace Length | Time Consuming (s) |
|---|---|---|---|---|
| SNN-Kalman | 104 | 64.89 | 654 | 0.6715 |
| MCSNG-SNN-K | 23 | 124.47 | 474 | 0.6081 |

*4.2. Testing by the Field Data*

In this part, we utilize the field data to test the performance of the proposed algorithm. The PBR field experiment is done with an uncooperative radar with frequency, PW and PRI agile, and it aims to detecting the air-flights. We make validations with the ADS-B (Automatic dependent surveillance-broadcast) dataset. The detection scope is from 80° to 170°. The detection range is from 50 km to 200 km. Due to unknown parameters of illuminator, the performance of pulse compression among several pulses degrades. To increase the detection probability, we reduce the CFAR rate to $10^{-2}$. After CFAR detection, we adopt the multi-beam amplitude comparison direction measurements. Figure 13a illustrates the final detection point map of the field data with 390 s duration. Two directions of jamming are located at 144° and 148°. Suppress the jamming in two directions and adopt the proposed clutter suppression algorithm. The structural element size is $5 \times 5$. The half number of the reference frames is set as 2. The suppression result is shown in Figure 13b. It is obvious that most of points are filtered out. Instead, points in three suspected track areas are retained. Referring to the ADS-B dataset, we plot the real-time civil flight information in Figure 13c, which is selected with the same duration and detection scope as the field data. The line with different colors stands for different flight track. There are three flights in the detection scope. Comparing with the ADS-B data, we can find that the proposed algorithm can effectively suppress the clutters and retain most of the target information.

Like the operations in Section 4.1.4, we test two compound tracking algorithms (MCSNG-NN and MCSNG-SNN-K) by the field data. The maximum velocity is set as 1200 m/s. The maximum accelerate velocity is set as 200 m/s². In Kalman filtering process, the maximum time period of blind prediction is set as 15 s. Each frame consists of the clustered point data with 0.5 s period. Figure 14 and Table 7 illustrates the tracing results and the performance indexes of NN-MHT and MCSNG-NN. Figure 15 and Table 8 illustrates the tracing results and the performance indexes of SNN-Kalman and MCSNG-SNN-K. Comparing with the ADS-B dataset, we marked the true tracks by the ellipses with dotted line. It is obvious that MCSNG-NN reduces the occurrence probability of false tracks relative to NN-MHT. In addition its time-consumption drops to 28.16% of the original NN-MHT time-consumption. Besides, similar conclusions are suitable to SNN-Kalman and MCSNG-SNN-K. We can find that MCSNG-SNN-K reduces the occurrence probability of false tracks and saves time relative to traditional SNN-Kalman.

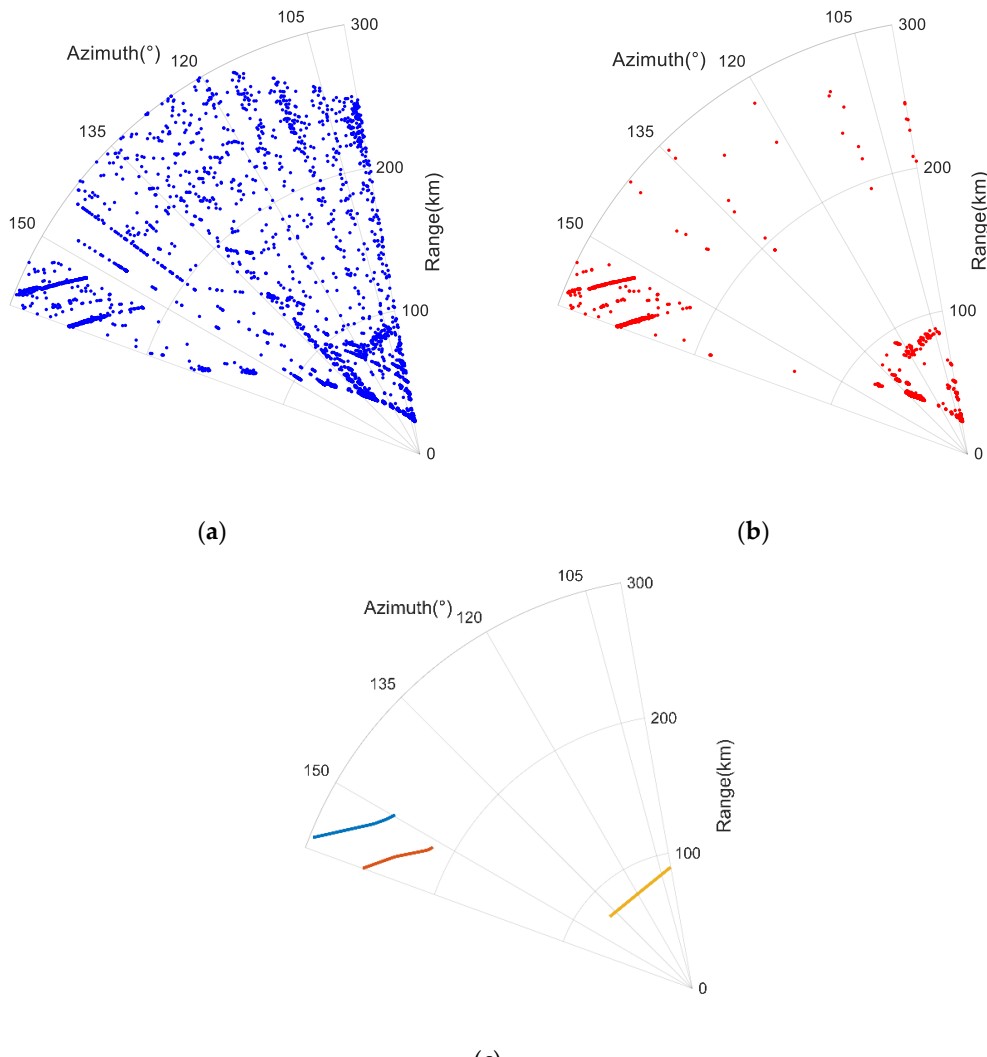

**Figure 13.** The contrast of before and after the suppression and the comparison with ADS-B dataset: (**a**) field data before suppression; (**b**) clutter suppression result; (**c**) real-time fights information from ADS-B dataset.

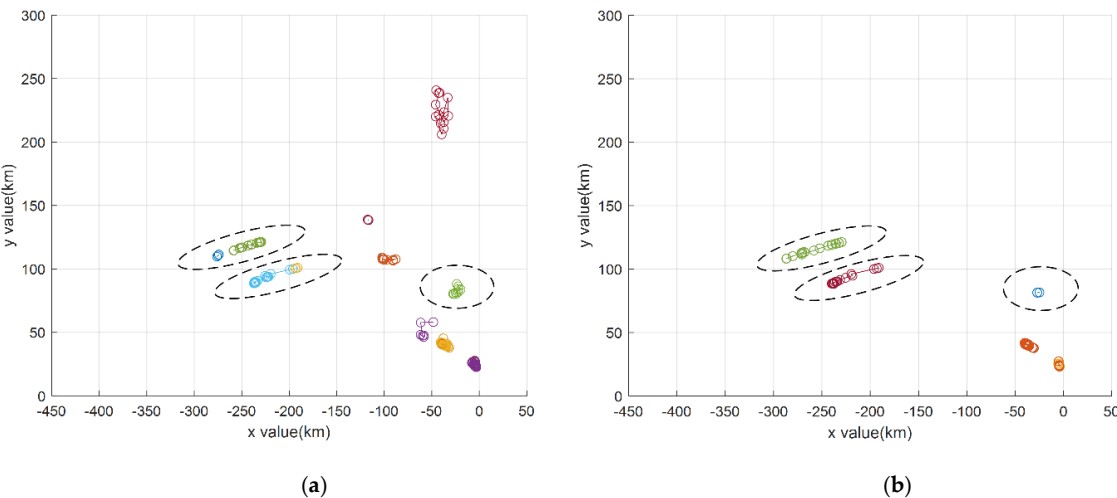

**Figure 14.** Tracing results of NN-MHT and MCSNG-NN: (**a**) NN-MHT tracing result; (**b**) MCSNG-NN tracing result.

**Table 7.** The performance indexes of NN-MHT and MCSNG-NN.

|  | Total Number of Traces | Mean Trace Length | Max Trace Length | Time Consuming (s) |
|---|---|---|---|---|
| NN-MHT | 31 | 38.7 | 135 | 4.83 |
| MCSNG-NN | 9 | 39.3 | 120 | 1.36 |

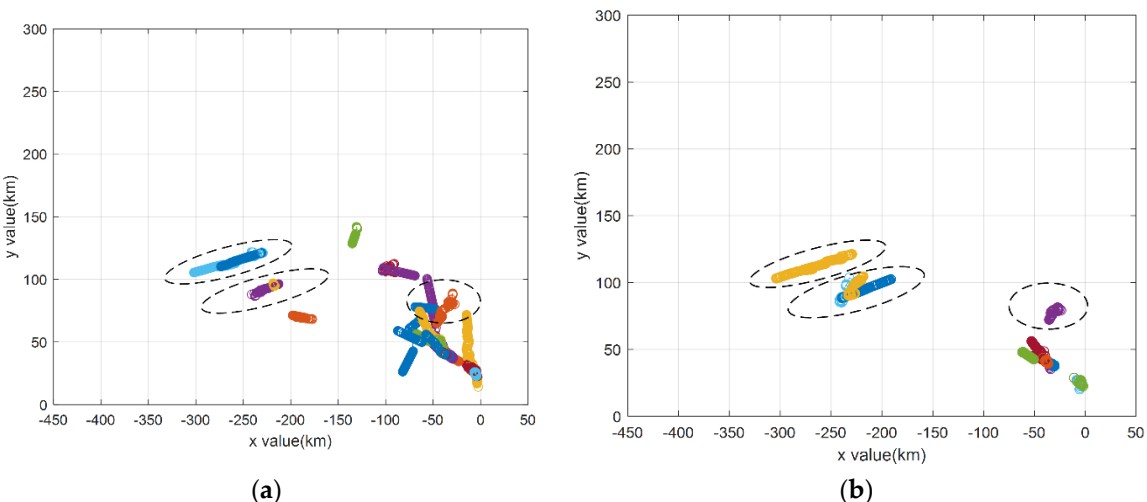

(**a**)　　　　　　　　　　　　　　　　　　　(**b**)

**Figure 15.** Tracing results of SNN-Kalman and MCSNG-SNN-K: (**a**) SNN-Kalman tracing result; (**b**) MCSNG-SNN-K tracing result.

**Table 8.** The performance indexes of SNN-Kalman and MCSNG-SNN-K.

|  | Total Number of Traces | Mean Trace Length | Max Trace Length | Time Consuming (s) |
|---|---|---|---|---|
| SNN-Kalman | 23 | 254.5 | 734 | 0.8536 |
| MCSNG-SNN-K | 13 | 196.6 | 424 | 0.6467 |

## 5. Conclusions

For PBR detection, especially for those illuminators with frequency, PRI and PW agile, it brings many challenges in following target tracing due to heavy clutters. Thus, combining with the features of PBR, a preprocessing operation is introduced before target tracing. In this paper, we propose a PBR cluttering suppression algorithm based on dilation morphology of non-uniform grid. According to the acquisition geometry of PBR, the nonuniform grid construction method is proposed at first. Then, iteratively separate false-alarm clutters from the point data based on dilation morphology. We perform experiments utilizing both simulated data and field data. Experiment results show that the proposed algorithm can effectively filter most false alarm clutters. Besides, combining with the tracing algorithm, it can enhance the PBR tracing performance, reduce the occurrence probability of false tracks and meanwhile save time. Furthermore, the theory of the proposed algorithm is also applicable for 3-D passive tracking, if the non-uniform grid for dilation is modified into cube. In current algorithm, the non-uniform grid is calculated through the first order Taylor expansion. Its reminder term is larger compared to the one of the higher order Taylor expansion. To balance the computation cost and the model accuracy, it is meaningful to exploit the maximum acceptable magnitude of measurement error in different positions. Future researches will focus on building up more specific non-uniform grid for clutter suppression combined with the target tracing, especially for the grids close to the baseline direction.

**Author Contributions:** Q.Z. and T.L. implemented the methods and designed the experiment. Q.Z. and J.P. performed the experiments and analyzed the data. Q.B. supervised the research. Q.Z. wrote the paper. All authors of the article provided substantive comments.

**Funding:** This research received no external funding.

**Conflicts of Interest:** The authors declare no conflict of interest.

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
