# Peer review of "PBR Clutter Suppression Algorithm Based on Dilation Morphology of Non-Uniform Grid"

_electronics, doi:10.3390/electronics8060708_

Round 1

Reviewer 1 Report

The objective of this paper is to reduce the clutter points by means dilation morpohology over a non-uniform grid to improve the performance of PBR (passive bi-static radar). Authors indicates that after CFAR processing (constant false alarm rate) a considerable false alarm clutter points appear that make worse the tracing.  For this fact, authors propose reduce this clutter points by means dilation morphology over a non-uniform grid.

Section 1 makes a review about passive and non-collaborative radars. The objectives are also clearly explained in this section. Section 2 describes the non-uniform grid and how it is constructed by means detection model which depends on the range and the angle. Section 3 describes the algorithm itself: dilatation operation, frame interrelation… and the final data. In Section 4, a computational analysis is done but this section needs additional review and additional clarification as the reviewer has detected some possible mistakes. Finally, the results are described in Section 5. Authors test their algorithm with simulated data and real data. They also add a test over two tracking algorithms.

· Authors support their algorithm with results from simulation and real data. The results confirm the improvement of the whole system.

·  All the stages and the basics of the algorithm are well and detailed described.

·  The bibliography is wide and supports the work

Some aspects to review:

·  Section 4, “Computation analysis”, is not clear for the reviewer.

o   The reviewer kindly suggests to review this short section. It should be explained better or even take this section out as no study about computational cost is followed up.

o   Additionally, it seems that some misunderstanding / swaps between additions and multiplications terms have been carried out. Please, check it. 

o   What does the operator o represent?

o   Why the k variable appears in the total amount of operations when it is defined as the frame number. It has not sense, unless that it represents the total number of frames for the processing. Please, clarify it.

o   What is the utility or the conclusion of this section? 

· The reviewer is not a native English speaker, so he is not confident to suggest any English correction, but he recommend to review English as some mistakes have been found: line 249, line 303 to 309,… It seems that sentences are not well structured.

·  For the reviewer is not clear what is the influence of the time interval between pulses. What is the problem that it is not a constant interval? How does this condition the results? Figure 7 does not provide useful information.

·  Figure 9 and suppression rates (line 279) are not clear for the reviewer. He does not understand why the authors say a decline rate of 10% but in the figures, the decline rate seems much greater. What was the detection accuracy rate before the suppression algorithm? Line 279. Author say that “the false alarm decline rate is the ratio of the number of the false points extracted to the whole number of the clutter points set before”.  In Table, this number is 10%. Then author say that “near 90% of clutter points are suppressed”.

·  Figure 10. Why the missing rate increase when the SCR also increases?

·  Figure 11. Whay the authors use this representation instead of the angle vs range? It is clearer but they should explain this change.  Additionally, authors could put the axis in km instead of m.

·  Table 5. Why does target 5 present a greater error? Is it due because it is closer to 180º? Are these targets in the same position than the previous ones?

·  Reviewer suggests to fusion Table 4 with Table 6 and Table 7 with Table 7 as it will save space and it will help the comparison between algorithms

·  The conclusions are a little short.

Some misspellings

· Line 145. There is an extra dot in “Figure.3.”

·  Sometimes the authors write the acronym before and the definition between brackets. In other cases, they do it the other way around. Please, uniform this

·  Lines 223 225. Shouldn’t mathematics variables be italic? Check it because sometimes they are in italic and other times not.

·  Figure 7. The x label has a mistake. It should be t (s) instead of t/s

·   Authors use percentage in text and then in Table 10 they used ratios between 0 and 1. They could use similar “units”

·   Suggestion. Put the units in the axis: Figure 8, Figure 9, Figure 14 and similar ones. It would be also fine if the authors use km units in the figures instead of m.

·  Figure 13. The first word “The” is bold and it should be plain.

·  Table 6 and Table 8. It seems that the title of the line 2, column 1 has a mistake. It should be MCSNG-SNN-K instead of MCSNG-NN.

· It would be helpful if the authors indicate the number of target over the figure 8c.

Author Response

Thanks for your careful review. Your suggestions are constructive and meaningful. Here is the response for each point. (details in Word file attachment)

Point 1: Section 4, “Computation analysis”, is not clear for the reviewer.

o   The reviewer kindly suggests to review this short section. It should be explained better or even take this section out as no study about computational cost is followed up.

o Additionally, it seems that some misunderstanding / swaps between additions and multiplications terms have been carried out. Please, check it.

o   What does the operator o represent?

o   Why the k variable appears in the total amount of operations when it is defined as the frame number. It has not sense, unless that it represents the total number of frames for the processing. Please, clarify it.

o   What is the utility or the conclusion of this section?

Response 1:

Thanks for your kind suggestion. After discussion, we decide to modify this section and take it into experiment part.

We have checked through all additions and multiplications terms.

Generally, the operator O(*) is used to evaluate the  computation amount of the algorithm. The time consuming is different, when the algorithm operates on different computers. So the computation evaluation index is not time but the logical term O(*). The prime characteristic of the operator O(*) is that it only retains the part with highest increasing rate and omits the constant coefficient. For example, assume .

Sorry, it is our mistakes in expression. The k variable represents the total number of frames for the processing.

The prime goal of the proposed algorithm is to reduce the computation amounts of tracing by suppressing the clutter points.

During tracing, the calculation amount relating to the number of clutter points is mostly caused by the Euclidean distance calculation between points. Assume M is the total number of points for tracing in each frame. In two consecutive frames, there are  point pairs for processing. For each point pair, the calculation of the Euclidean distance needs 3 additions and 3 multiplications. Thus, the calculation amount of the Euclidean distance is  additions and  multiplications. Where k is the total number of frames.

Suppose our algorithm can suppress 90% clutter points. And this suppression process only consumes o(k  M) additions and o(k) multiplications. After suppression, the calculation amount of the Euclidean distance will descend to the 1% of the original. Thus, we pay low calculation amounts for reducing much more computation amounts of tracing.

Point 2: The reviewer is not a native English speaker, so he is not confident to suggest any English correction, but he recommend to review English as some mistakes have been found: line 249, line 303 to 309,… It seems that sentences are not well structured.

Response 2: Thanks for your kind suggestion.

Line 249:

Table 2 lists the track settings of five targets.  

Line 303 to 309:

Combine the clutter suppression with two tracking algorithms mentioned above respectively. For each frame, the tracking process follows with the clutter suppression in pipeline operation. Based on the difference of the tracking process, we name these compound algorithms as MCSNG-NN and MCSNG-SNN-K respectively. To make comparison, we utilize the original data without clutter suppression for tracing. Choose four indexes (total number of traces, mean trace length, maximum trace length, time consuming) to evaluate the tracing performance. Set the maximum velocity and the maximum accelerated velocity as 1000 m/s and  respectively.

Point 3: For the reviewer is not clear what is the influence of the time interval between pulses. What is the problem that it is not a constant interval? How does this condition the results? Figure 7 does not provide useful information.

Response 3: The time interval between pulses represents the time interval between frames. Due to random time interval, the target point in consecutive frames moves with random spacing.

Generally, modern radar with phased array transmits the agile signal for low probability of intercept. Especially, the PRI (pulse repetition interval) of signal is agile. And the background of our research is to explore the passive radar employing non-cooperative radar as illuminator.

The random time interval does not directly condition the results. It just increases the difficulty of traditional tracing. Besides, this may result in the appearance of breaking tracks. Section 5.1 is the experiment by simulated data. To make the simulated scenario closer to reality, Figure 7 shows the random interval allocation between adjacent pulses.

Point 4: Figure 9 and suppression rates (line 279) are not clear for the reviewer. He does not understand why the authors say a decline rate of 10% but in the figures, the decline rate seems much greater. What was the detection accuracy rate before the suppression algorithm? Line 279. Author say that “the false alarm decline rate is the ratio of the number of the false points extracted to the whole number of the clutter points set before”.  In Table, this number is 10%. Then author say that “near 90% of clutter points are suppressed”.

Response 4:  

Figure 9 is the contrast figure of before and after suppression. Figure 9(a) is the data from CFAR processor. Figure 9(b) is the clutter suppression result. It is clear that most of clutter points are suppressed.

Sorry, we did not mention the concept of the suppression rate in this paper(line 279. Maybe, the concept you mean to ask is the false alarm decline rate. It is the ratio of the number of clutter points retained and the total number of clutter points set before. For example, there are 100 clutter points set in the simulated data. After the clutter suppression, 90 clutter points are removed. The suppression rates here is 10%.

Table 3 is corresponding to Figure 9. And the decline rate of 10% is the performance index value of the data from Figure 9, where SCR is set as -1.26 Db.

The detection accuracy rate is the ratio of the number of target points correctly extracted to the total number of target points set before. For example, there are 100 target points set in the simulated data. After the clutter suppression, only 90 target points are retained. The detection accuracy rates here is 90%.

The definition of the false alarm decline rate is the ratio of the number of clutter points retained and the total number of clutter points set before. For example, there are 100 clutter points set in the simulated data. After the clutter suppression, 90 clutter points are removed. The suppression rates here is 10%. In Table, the false alarm decline rate is 10%. Thus, the number of clutter points suppressed is (1-10% = 90%) 90% of the original number.

Point 5: Figure 10. Why the missing rate increase when the SCR also increases?

Response 5:  It is a good question. The basic theory of our proposed method is to extract the true points with the help of the location dependency among track points. In low SCR, there are massive clutter points in surveillance area. So, as the SCR decreases, the number of clutter points distributing around the true points increases. On one hand, it results in erroneous judgement of these clutter points. On the other hand, these false alarm points can contribute to detecting those true points with large measurement error. This can make up the deficiency of the mask. Thus, when the SCR increases, the missing rate also slowly increases. However, thanks to the important role of the mask, the missing rate is relatively stable, when the SCR exceeds some value.

Point 6: Figure 11. Why the authors use this representation instead of the angle vs range? It is clearer but they should explain this change.  Additionally, authors could put the axis in km instead of m.

Response 6: In current tracing algorithm, the operations are done in cartesian coordinates. Especially, it is convenient for the calculation of the Euclidean distance. Besides, Kalman prediction is also done in cartesian coordinates. So the results are present in cartesian coordinates.

Thanks for your suggestion. We change the axis into km in the revision edition.

Point 7: Table 5. Why does target 5 present a greater error? Is it due because it is closer to 180º? Are these targets in the same position than the previous ones?

Response 7: Because target 5 is closer to 180º, the direction of the baseline. From Figure 8(c), we can find that the error of measurement is greater when it is closer to 180º. These targets are in the same position than the previous ones. All targets in section 5.1 are in the same position.

Point 8: Reviewer suggests to fusion Table 4 with Table 6 and Table 7 with Table 7 as it will save space and it will help the comparison between algorithms.

Response 8: Thanks for your suggestion. But I think Table 4 and Table 6 have no relationship, even though the performance indexes in both are same. Table 4 and Table 6 are the tracing results in different parameter sets. They have no comparability. I am afraid that fusion of  Table  4 and Table 6 may result in some misunderstandings for readers. Similar reasons also apply to Table 7 and Table 8.

Point 9The conclusions are a little short.

Response 9: Thanks for your suggestion. We will expand the conclusion in the revision edition.

Here is the modified conclusion. The added content is underlined.

5. Conlusion

For PBR detection, especially for those illuminators with frequency, PRI and PW agile, it brings many challenges in following target tracing due to heavy clutters. Thus, combining with the features of PBR, a preprocessing operation is introduced before target tracing. In this paper, we propose a PBR cluttering suppression algorithm based on dilation morphology of non-uniform grid. According to the acquisition geometry of PBR, the nonuniform grid construction method is proposed at first. Then, iteratively separate false-alarm clutters from the point data based on dilation morphology. We perform experiments utilizing both simulated data and field data. Experiment results show that the proposed algorithm can effectively filter most of false alarm clutters. Besides, combining with the tracing algorithm, it can enhance the PBR tracing performance, reduce the occurrence probability of false tracks and meanwhile save time-consuming. Furthermore, the theory of the proposed algorithm is also applicable for 3-D passive tracking, if the non-uniform grid for dilation is modified into cube.  In current algorithm, the non-uniform grid is calculated through the first order Taylor expansion. Its reminder term is larger comparing to the one of the higher order Taylor expansion. To balance the computation cost and the model accuracy, it is meaningful to exploit the maximum acceptable magnitude of measurement error in different positions. Future researches will focus on building up more specific non-uniform grid for clutter suppression combined with the target tracing, especially for the grids close to the baseline direction.

Point 10:

Line 145. There is an extra dot in “Figure.3.”

Sometimes the authors write the acronym before and the definition between brackets. In other cases, they do it the other way around. Please, uniform this.

Lines 223 225. Shouldn’t mathematics variables be italic? Check it because sometimes they are in italic and other times not.

Figure 7. The x label has a mistake. It should be t (s) instead of t/s

Authors use percentage in text and then in Table 10 they used ratios between 0 and 1. They could use similar “units”

Suggestion. Put the units in the axis: Figure 8, Figure 9, Figure 14 and similar ones. It would be also fine if the authors use km units in the figures instead of m.

 Figure 13. The first word “The” is bold and it should be plain.

 Table 6 and Table 8. It seems that the title of the line 2, column 1 has a mistake. It should be MCSNG-SNN-K instead of MCSNG-NN.

 It would be helpful if the authors indicate the number of targets over the figure 8c.

Response 10: Thanks for your careful check. We will amend in the revision edition.

Reviewer 2 Report

Please refer to the attach.

Author Response

Thanks for your careful review. Your suggestions are constructive and meaningful. Here is the response for each point. (details in the Word file attachment)

Point 1: line 14: CFAR is firstly described here, so the full explanation should be located

here.

Response 1:

The full explanation of CFAR is constant false alarm. We will amend in the revision edition.

Point 2: 

l line 21: the algorithm proposed → the proposed algorithm

line 136: as (9)(10) shown → shown in Eqs. (9) and (10)

line 145: 100° to 160° and from 20km to 100km → 100 ° to 160 ° in azimuth and
20 km to 100 km in range.

line 177: 13) shown → shown in (13)

line 192: the algorithm proposed → the proposed algorithm

Figure 8: (d) Detection results → (d) detection results

Response 2: Thanks for your careful check. We will amend in the revision edition.

Point 3:  line 52-64: Authors described the new challenges for improving the performances of PBR. Among these challenges, authors had better mention about what items will be dealt with this research among them.

Response 3:

Thanks for your kind suggestion.

But it can be found that all the mentioned items have been dealt with this research:

The space synchronization accuracy results in poor location precision. So, we calculate bi-static error model in polar grid construction for PBR.

Increasing redundant data results in huge computation of tracing. So, this research adopts the grid-based method to avoid calculating the Euclidean distance of point pairs. This can release computation burden in dense data environment.

The degraded pulse compression performance leads to failing detection of target in some echo pulses. So, in section 3.2, we utilize several consecutive reference frames to ensure true target points.

The illuminator parameters are agile pulse by pulse. So, the interval between adjacent frames are random. In section 3.2, we utilize a rectangular structural element B with reasonable size to separate false alarm clutter.

Increasing false-alarm rate results in lots of false alarm clutter points. It will disturb the target tracking. And the prime goal of our algorithm is to suppress clutter points as far as possible with low computation load.

Point 4: Figure 3: It is very hard to recognize blue and red lines. And the units of range and
azimuth would be added in the Figure.

Response 4: To make it easier to distinguish colours, we add a partial enlarged view in Figure 3. And we add the units of range and azimuth in the Figure 3.

Point 5: line 183: What does “curse estimation” mean?

Response 5:  Sorry, it is our spelling mistake. It should be “coarse estimation”.

Point 6: line 229 – 241: I think these descriptions would be placed into Chapter 5.

Response 6:  Thanks for your kind suggestion. After our discussion, we decide to place the content of section 4 into section 5.1.

Point 7:  Table 2: The analyzed domain looks range and azimuth in Figures of the

manuscript. Start position in Table 2 is described by range and range. For better

understanding, the start position would be explained by range and azimuth.

Response 7: Thanks for your kind suggestion. We will add one column to explain start positions in polar coordinates in Table 2.

Start position (km, degree)

in polar coordinates

Start position (km) 

in cartesian coordinates

Track Slope

Track intercept

(km)

Velocity (m/s)

Target 1

(86.023,144.5)

(-70, 50)

5

60

800

Target 2

(70.456,96.5)

(-8, 70)

10

100

-750

Target 3

(80.623,82.9)

(10, 80)

-3

10

600

Target 4

(76.158,113.2)

(-30. 70)

-10

80

400

Target 5

(70.711,135)

(-50, 50)

30

55

-600

Point 8: Line 275 : The mask size is 3×3 km?

Response 8: No, here the unit is the number of grids. The mask size is abstract not real. The real size of the mask is not fixed but various along with the location. As Figure below (Figure 5) shown, there are two structural elements with different real shape and real size. But both are regarded as the mask with same size (3×3) during processing.

Point 9: Line 277 : Three variables for examining the performance should be defined.

Response 9: 

In previous edition, these variables have been defined in line 278 to line 281.

line 278 to line 281:

The detection accuracy rate is the ratio of the number of extracted target points to the total number of the target points. The false alarm decline rate is the ratio of the number of the extracted clutter points to the total number of the clutter points set before. The miss detection rate is the ratio of the number of missing target points to the total number of  target points.

Point 10Figure 10: It is hard to recognize the line colors.

Response 10: Thanks for your suggestion. We will change original lines into bold lines in Figure 10.

Point 11: Figures 14 and 15: What does different colors in both Figures?

Response 11: In figure 14 and 15, we use different colors to distinguish different tracks in tracing results. Since color types in Matlab are limited, some tracks in Figure may be in same colors. But it does not cause a problem. One line segment stands for one track.

Reviewer 3 Report

44-45 If the hypothesis of a phased array is needed, their availability in zones of interest should be deepened.

113 It could be worth briefly explaining how Rs is estimated from only the echo channel

117-118 I presume M(.) is the non-uniform grid. Please explain or define M before using it.

120 Since RHO1 is a length and RHO2 is a squared length, I suggest you to change symbols accordingly for a clever explanation. For example you could define RHO2 squared ending up in DELTA_Rr = RHO1*DELTA_R + RHO2^2 * DELTA_THETA.

125 Rephrase to show that also DELTA_THETA is a error.

EQUATION 4 To be precise, the equal sign should be substituted with a approximation sign (≈) or a o(DELTA_r,DELTA_THETA) should be added as last term. Also all next math should be changed accordingly.

131 DELTA_R is linear in DELTA_r and DELTA_THETA only under the approximation of first order Taylor series. It could be worth to exploit the maximum acceptable magnitude of both - the maximum size of o(DELTA_r,DELTA_THETA) - to keep enough validity.

Author Response

Thanks for your careful review. Your suggestions are constructive and meaningful. Here is the response for each point. (details in following Word attachment)

Point 1: 44-45 If the hypothesis of a phased array is needed, their availability in zones of interest should be deepened.

Response 1:

Thanks for your suggestion. We have amended in the revision edition.

Here is our added content.

Comparing to the traditional mechanical scanning radar, the phased array has flexible multi-beam scanning and various beam dwell time. It can achieve search and tracing simultaneously. Nowadays, many modern radars are equipped with phased array for detection. Thus, it is meaningful to explore the phased array radar signal as the illuminator.

Point 2: 113 It could be worth briefly explaining how Rs is estimated from only the echo channel

Response 2:  In line 113, we do not mention “only” in the sentence. Maybe, our expression confused you.

At first, complete the time synchronization of the echo channel with the help of the direct wave channel. Then, measure the difference between  and L through the time delay of echo. Since L is fixed,  can be directly measured from echo channel. 

Point 3: 117-118 I presume M(.) is the non-uniform grid. Please explain or define M before using it.

Response 3:

Thanks for your reminder. M is just an arbitrary point located at position (R,). We will remove M and retain only the coordinates to avoid ambiguity.

Point 4: 120 Since RHO1 is a length and RHO2 is a squared length, I suggest you to change symbols accordingly for a clever explanation. For example you could define RHO2 squared ending up in DELTA_Rr = RHO1*DELTA_R + RHO2^2 * DELTA_THETA.

Response 4: Sorry, we made some mistakes when typing the equation (5).

Eq.(5) should be as below equation shown.

Thus, ,  .

But we do not think  is a squared length.

Point 5: 125 Rephrase to show that also DELTA_THETA is a error.

Response 5:  Thanks for your careful check. It is our mistake.

It should be “the azimuth angle error

Point 6: line 229 – 241: I think these descriptions would be placed into Chapter 5.

Response 6:  Thanks for your kind suggestion. After our discussion, we decide to place the content of section 4 into  section 5.1.

Point 7: EQUATION 4 To be precise, the equal sign should be substituted with a approximation sign (≈) or a o(DELTA_r,DELTA_THETA) should be added as last term. Also all next math should be changed accordingly.

Response 7: Thanks for your reminder.  We add the  in Equation (4)(5) and change the equal sign into an approximation sign in Equation (6).

Point 8: 131 DELTA_R is linear in DELTA_r and DELTA_THETA only under the approximation of first order Taylor series. It could be worth to exploit the maximum acceptable magnitude of both - the maximum size of o(DELTA_r,DELTA_THETA) - to keep enough validity.

Response 8:

Thanks for your constructive suggestion.

For first order Taylor series expansion of (2) at position:

       (4)

The last part of the Equation (4) is the Lagrange remainder term.

Assume the  as the remainder term.

Because the second order partial derivative of  is continuous. Its abstract must have an upper bound in the neighbourhood of . 

Suppose is the upper bound value of   in the neighbourhood of  within  and .is the upper bound value of   in the neighbourhood of  within  and .is the upper bound value of   in the neighbourhood of  within  and .

M is the maximum abstract value of , and .

Then, 

Assume , 

Thus, , . ().

So,

Assume ,  

Then,

Thus,  .

Suppose the maximum acceptable magnitude of the remainder term is T.

To some extent,  can reflect the maximum acceptable magnitude of both   and  together. So, to keep enough validity, the maximum acceptable magnitude  should be . 

 It is a complex mathematical problem to derive the formula expression of  , which is beyond our current abilities. Thus, we will exploit this issue in the future work.

Round 2

Reviewer 2 Report

I would like to recommend that the modified manuscript would be accepted in present form.